# Towards a transcriptomic biomarker for the classification of melanocytic neoplasms

**Elizabeth S. Borden**[1,2], **Colin T. Hastings**[1], **Nithish Prakash**[1], **Tyler Kuo**[1], **Edgar Tapia**[3], **Michael Yozwiak**[3], **Paul Sagerman**[4,5], **Danielle Vargas de Stefano**[6], **Kenneth H. Buetow**[7,8], **Melissa A. Wilson**[7,8,9], **Clara Curiel-Lewandrowski**[3,4], **Hsiao-Hui Sherry Chow**[3,4], **Bonnie J. LaFleur**[10,11], **Karen Taraszka Hastings**[1,2,3]*

1 Department of Dermatology, University of Arizona College of Medicine-Phoenix, Phoenix, Arizona, United States of America, 2 Phoenix Veterans Affairs Health Care System, Phoenix, Arizona, United States of America, 3 University of Arizona Cancer Center, University of Arizona, Tucson, Arizona, United States of America, 4 Department of Medicine, University of Arizona College of Medicine Tucson, Tucson, Arizona, United States of America, 5 Sagis Diagnostics, Houston, Texas, United States of America, 6 Department of Pathology and Laboratory Medicine, Phoenix Children's Hospital, Phoenix, Arizona, United States of America, 7 School of Life Sciences, Arizona State University, Tempe, Arizona, United States of America, 8 Center for Evolution and Medicine, Arizona State University, Tempe, Arizona, United States of America, 9 Comparative Genomics and Reproductive Health Section, Center for Genomics and Data Science Research, National Human Genome Research Institute, National Institutes of Health, Bethesda, Maryland, United States of America, 10 BIO5 Institute, University of Arizona, Tucson, Arizona, United States of America, 11 R. Ken Coit College of Pharmacy, University of Arizona, Tucson, Arizona, United States of America

* khasting@arizona.edu

## Abstract

Histopathologic diagnosis of thin, invasive cutaneous melanoma (CM) is only 34–62% accurate. Therefore, we sought to develop a transcriptomic biomarker to distinguish benign from malignant melanocytic neoplasms. We generated a targeted RNA-Sequencing dataset (TempO-Seq) of benign nevi (BN; n = 50) and CM (Breslow depth ≤ 1.0 mm; n = 51) and demonstrated enrichment of immune-related pathways among the 450 differentially expressed genes. Next, we trained a putative transcriptomic biomarker in two datasets, including BN and CM, and one dataset with CM in association with a nevus, macrodissected into CM and nevus regions. We refer to the nevus portion of CM in association with a nevus as progressing nevi (PN), since these nevi progressed to CM. Principal component analysis showed that PN samples clustered in a component intermediate to BN and CM. Ordinal regularized regression selected *PYGL, AP000845.1, PHYHIP, WSCD1, FBXO7, TRPM1, SLC4A4, NALCN, FRMD4B, HHATL, COL1A1, CRYM, EPOP, RGS1, KRT6C, IGHG1, CNTN1, MMP11, GZMM, AP001880.1, TTYH3, TMEM132A,* and *PRAME*; these genes were consistently selected in 1000 models using data from bootstrap resamples and had a single model predictive accuracy of at least 0.90 (area under the receiver operator characteristics curve). Linear regression models fit with these 23 genes in the TempO-Seq data, and publicly available microarray datasets from BN, dysplastic nevi, and CM, showed high consistency in the magnitude and directionality of gene expression

**Data availability statement:** The full TempO-Seq expression dataset acquired for this manuscript and the related metadata are provided as supplementary information tables. Expression data from the published datasets are available through public repositories: the Badal, Schartl, Krueger, and Scatolini datasets are publicly available through the Gene Expression Omnibus (GEO) with the following accessing numbers: GSE98394, GSE112509, GSE114445, GSE12391. The Bastian dataset is available as a controlled access dataset through the database of Genotypes and Phenotypes (dbGaP) with accession number phs001550.v2.p1l (https://www.ncbi.nlm.nih.gov/projects/gap/cgi-bin/study.cgi?study_id=phs001550.v2.p1).

**Funding:** This research was supported by the National Institutes of Health/National Cancer Institute grants UG1CA242596 (HHSC, KTH, CCL, BJL) and F30CA281056 (ESB). This research benefited from support for related projects by the Merit Review Award I01-BX005336 from the United States Department of Veterans Affairs (VA), Biomedical Laboratory Research and Development Service (KTH) and by the Intramural Research Program of the National Human Genome Research Institute, National Institutes of Health (MAW). The contents do not represent the views of the VA or the United States Government. The funders had no role in study design, data collection and analysis, decision to publish, or preparation of the manuscript.

**Competing interests:** I have read the journal's policy and the authors of this manuscript have the following competing interests: E.S.B., B.J.L., and K.T.H. have a provisional patent on work contained herein (US Serial No. 63/885,019). E.T. is currently an employee of Caris Life Sciences. B.J.L. is an independent contractor for MDeloris Medical Systems, a consultant for Linshom Medical, and a consultant for iNanoBio, Inc, a member of the Data Safety Monitoring Board for the National Institutes of Health, Nutrition for Precision Health. C.C.-L. receives royalties from UpToDate. All other authors have no conflicts to disclose.

differences between nevi and CM. Furthermore, immunohistochemical staining showed consistent protein-level changes in MMP11 and PYGL. These results illuminate the potential for a transcriptomic biomarker to differentiate benign from malignant melanocytic neoplasms and improve the accuracy of melanoma diagnosis.

## Author summary

The diagnosis of thin melanoma is an ongoing clinical challenge. Therefore, our goal was to develop a model from gene expression data that could distinguish nevi from melanoma. We began by acquiring a new dataset and demonstrating that the differences between benign nevi and melanoma were predominated by changes in immune-related genes. Subsequently, we used three publicly available datasets to create a model that distinguishes benign nevi, melanoma, and nevi that progress to melanoma. We demonstrate that the resulting model has high predictive accuracy. Furthermore, the genes selected for the model show consistent changes between nevi and melanoma across multiple existing datasets from different technologies. Overall, these results provide the first steps in the creation of a clinically useful model for distinguishing melanoma from related noncancerous lesions.

## Introduction

Invasive cutaneous melanoma (CM) is the fifth most common cancer in both men and women in the United States [1]. The incidence of CM continues to increase despite public health initiatives promoting sun protection, with the increase primarily driven by early, thin CM [2,3]. The American Cancer Society estimates that there will be 100,640 new cases of invasive CM and at least 99,700 cases of melanoma in situ in 2024 [1]. Therefore, CM continues to be a significant health challenge in the United States.

Pathologist diagnosis of melanocytic neoplasms, specifically for thin T1a (<0.8 mm) CM, is neither reproducible nor accurate, with 20% variability of over- and underdiagnosis [4–6]. Fellowship training in dermatopathology and a second opinion can improve diagnostic accuracy but does not eliminate misclassification [5]. The misclassification rate by dermatopathologists is highest for dysplastic nevi (DN) with moderate or severe atypia, melanoma in situ, and T1a CM, ranging from 44-69% [5]. Features of dysplastic nevi include asymmetry, junctional melanocytic proliferation concentrated along the sides and tips of rete ridges, peripheral extension of the epidermal component beyond the dermal component, elongation and bridging of rete ridges, lamellar fibroplasia in the papillary dermis, and chronic inflammation in the papillary dermis. In thin melanoma there is confluent lentiginous junctional melanocytic proliferation involving suprapapillary plate regions, pagetoid intraepidermal scatter of melanocytic cells, and cytologic atypia of melanocytic cells. Overdiagnosis of

CM leads to increased procedures, risks, and costs without benefit, and underdiagnosis leads to the potential for metastasis. Thus, there is a clinical need for improved tools to aid the diagnosis of melanocytic neoplasms ranging from DN with moderate atypia to thin T1a CM [7,8].

Molecular technologies have advanced significantly over the past decade, leading to various efforts to improve CM diagnostic accuracy. Among these, immunohistochemistry (IHC) for detecting the melanoma-associated antigen PRAME (Preferentially Expressed Antigen in Melanoma), gene expression profiling [9–12], and tests that identify chromosomal aberrations in neoplastic melanocytes (tumor cytogenetics)—such as fluorescence in situ hybridization [13–16], and array comparative genomic hybridization/single nucleotide polymorphism arrays [17]—have been implemented with varying degrees of adoption in clinical practice. Among these technologies, the most widely used discriminatory test is the MyPath Melanoma assay (Castle Biosciences, Phoenix, AZ), a 23-gene expression profile designed to provide an objective, reproducible, and accurate adjunctive tool for distinguishing CM from benign nevi (BN) [9–12]. This test is particularly intended for dermatopathologists evaluating primary cutaneous melanocytic neoplasms where the diagnosis of CM versus BN remains uncertain. While some studies suggest that the test improves diagnostic confidence and reduces unnecessary procedures, a recent real-world validation study comparing MyPath results with final dermatopathologist-rendered diagnoses showed lower sensitivity and specificity than previously reported [18]. As molecular technologies continue to evolve and larger, more diverse sample cohorts—including variations in age, gender, and clinical-histological characteristics—become available, further studies are needed to refine and enhance melanoma diagnostic accuracy.

Here, we created a new dataset of BN, DN, and CM and probed the biological changes that occur during melanomagenesis, adjusting for differences in patient age and evaluating for sex differences. We then combined publicly available datasets: two from BN and CM, and one from CM in association with a nevus, which were macrodissected into CM and nevus regions. We refer to the nevus portion of a CM in association with a nevus as progressing nevi (PN) for nevi known to progress to CM. We created a transcriptomic biomarker that differentiates BN, PN, and CM and demonstrated that the genes selected by this model are robust across datasets and technologies.

## Results

### Batch effect identified in the TempO-Seq data likely due to length of specimen storage

To create a transcriptomic biomarker to distinguish BN, DN, and CM, TempO-Seq RNA expression data was obtained for 50 BN samples and 51 CM samples archived through a community pathology lab, as well as 66 samples from clinically atypical nevi (AN, resulting in histopathologic diagnosis of 61 DN and 5 BN samples) from a prior clinical trial [19]. TempO-Seq technology was selected due to the higher throughput and lower cost compared with traditional RNA sequencing (RNA-Seq) technology.

We began by examining the demographic characteristics of the acquired samples (Table 1). The CM samples were from significantly older patients than the BN ($p = 4.2 \times 10^{-7}$, Wilcoxon rank sum test) and AN ($p = 1.7 \times 10^{-7}$, Wilcoxon rank sum test), while there was no significant difference in patient age between BN and AN ($p = 0.22$, Wilcoxon rank sum test). The sex distribution was significantly different for BN compared to CM ($p = 0.0024$, chi-squared test) and significantly different for BN compared to AN ($p = 0.020$, chi-squared test), but not significantly different for AN and CM ($p = 0.17$, chi-squared test). The site distribution was significantly different for BN compared to AN ($p = 0.0086$, chi-squared test) and for AN compared to CM ($p = 5.5 \times 10^{-7}$, chi-squared test), but not significantly different for BN compared to CM ($p = 0.062$, chi-squared test).

Visualization of the TempO-Seq expression data with a principal components analysis (PCA) demonstrated clustering of all BN and CM samples from the community pathology lab separately from the AN samples from the University of Arizona (UA) or Stanford University, despite processing of formalin-fixed, paraffin-embedded (FFPE) samples for TempO-Seq at the same time (Fig 1A). Notably, the FFPE blocks from the UA and Stanford had been stored for a longer time than

**Table 1. Distribution of demographic characteristics for benign nevi (BN), atypical nevi (AN), and cutaneous melanoma (CM) samples subjected to TempO-Seq.**

|  | BN | AN | CM | p-values |
|---|---|---|---|---|
| **Number of patients** | 46 | 40 | 45 | -------------- |
| **Number of samples** | 50 | 66 | 51 | -------------- |
| **Patient age in years**[1] | Mean 43.33 ± SD 17.21 Range: 13 – 77 Median: 41.5 | Mean 46.75 ± SD 9.15 Range: 27–67 Median: 48 | Mean 63.73 ± SD 13.86 Range: 32 – 97 Median: 66 | BN vs. AN: 0.22 BN vs. CM: **4.2 x 10⁻⁷** AN vs. CM: **1.7 x 10⁻⁷** |
| **Sex**[1] | Female 31/46 (67%) Male 15/46 (33%) | Female 16/40 (40%) Male 24/40 (60%) | Female 15/45 (33%) Male 30/45 (67%) | BN vs. AN: **0.020** BN vs. CM: **0.0024** AN vs. CM: 0.17 |
| **Site** | Head/neck 7/50 (14%) Upper extremity 5/50 (10%) Trunk 32/50 (64%) Lower extremity 6/50 (12%) | Head/neck 0/66 (0%) Upper extremity 6/66 (9.1%) Trunk 56/66 (84.8%) Lower extremity 4/66 (6.1%) | Head/neck 11/51 (21.6%) Upper extremity 11/51 (21.6%) Trunk 19/51 (37.3%) Lower extremity 10/51 (19.6%) | BN vs. AN: **0.0086** BN vs. CM: 0.062 AN vs. CM: **5.5 x 10⁻⁷** |
| **Histologic information** | Compound 50/50 (100%) | Dysplastic Junctional 9/66 (13.6%) Dysplastic Compound 52/66 (78.8%) Benign Compound 5/66 (7.6%) | Superficial spreading melanoma 51/51 (100%) Breslow thickness Mean 0.44 +/- S.D. 0.25 mm | -------------- |

[1]Age and sex were calculated across patients rather than sample.

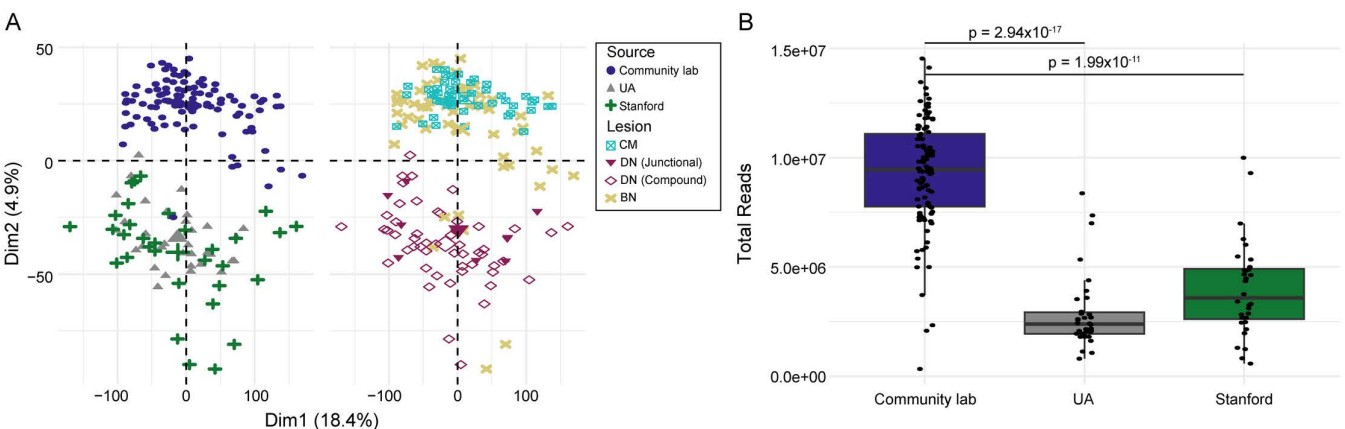

**Fig 1. Batch effect identified in the TempO-Seq data likely due to length of specimen storage. A)** Principal component analysis (PCA) on TempO-Seq expression data from benign nevi (BN; n = 50) and cutaneous melanoma (CM; n = 51) from a community pathology lab and dysplastic nevi (DN) and BN from the University of Arizona (UA; n = 31 DN, n = 3 BN) or Stanford University (n = 30 DN, n = 2 BN). PCA plots are colored by the site of formalin-fixed, paraffin-embedded processing (left) and lesion type (right). **B)** Comparison of the total number of sequencing reads from the specimens processed by the community pathology lab, UA, and Stanford. A Kruskal-Wallis test with Dunn's multiple comparison test was used to examine scale differences.

the FFPE blocks from the community pathology lab. The BN and CM samples from the community pathology lab were archived between 2017 and 2021, whereas the AN samples from the UA and Stanford were archived between 2009 and 2010. Consistent with the older age of the samples from the UA and Stanford, the total read count in these samples was significantly lower than that of the more recently archived CM and BN specimens (Fig 1B). Because the majority of the BN and all of the CM samples came from the community pathology lab and all of the DN samples came from UA or Stanford,

the observed batch effect was confounded with the lesion type. For this reason, all subsequent analyses of the TempO-Seq data were restricted to the BN and CM samples.

## Differentially expressed genes between BN and CM are consistent in males and females

We performed differential expression and pathway enrichment analyses to examine the biological differences between BN and CM. Previous work has demonstrated that both age and sex can impact the gene expression profiles of melanoma samples [20]. Therefore, to isolate the effects of the lesion type from the differences in the age and sex distributions between BN and CM, we performed the differential expression analysis separately for males and females and used age-adjusted expression values as described in the methods. While sun exposure has also been identified as influential in gene expression in normal skin [21,22] and has preliminary evidence of influencing gene expression in melanoma [23], no significant differences in the site were identified between BN and CM specimens. Thus, site was not considered as a covariate in the differential expression analysis. We identified 409 differentially expressed genes between BN and CM in females (236 up in CM vs. 173 up in BN, Fig 2A) and 106 differentially expressed genes between BN and CM in males (76 up in CM vs. 30 up in BN, Fig 2B). We then evaluated the overlap between the differentially expressed genes from males and females (Fig 2C). Across the comparisons within males and females, we detected 450 unique differentially expressed genes. Of these, 344 genes (76.4%, 193 upregulated and 151 downregulated in CM) were identified as significant solely in female samples, 41 genes (9.1%, 33 upregulated and 8 downregulated in CM) were identified as significant solely in male samples, and 65 genes (14.4%, 43 upregulated and 22 downregulated in CM) were identified as significant in both male and female samples.

We then evaluated whether the genes identified in the segregated analyses showed similar expression changes between males and females. We plotted the log fold change of the genes significant in males and females, males only, and females only (Fig 2D-F). The vast majority of genes demonstrated consistency in directionality and magnitude in males and females. However, eighteen genes were identified with log fold change differences of at least one between males and females (annotated in Fig 2D and 2F); some of these genes, *BASP1, HLA-A, HLA-DQB1, HLA-DRB5, MELK, S100A7, SERPINA1,* and *SOX9,* have been previously identified as being associated with sex differences, interacting with the estrogen receptor, or being regulated by estrogen [24–31]. While six X-linked genes were identified as differentially expressed in BN vs. CM (*RPS4X* in males and females, *FLNA* in males only, and *SASH3, GPR143, CYBB, and TCEAL4* in females only) all of these genes had a consistent direction of change between males and females with a log fold change difference of less than one.

To investigate whether sex differences in the 18 genes between BN and CM were detected in another dataset, we performed the same analysis in the Schartl dataset (Table 2). Of the three RNA-Seq datasets, the Bastian dataset did not have BN samples available and the Badal dataset had no BN samples from male patients, preventing the analysis in these datasets. Within the Schartl dataset, the majority of the 18 genes had a similar log fold change in males and females, and five genes had a log fold change difference of at least 1 between males and females in the opposite direction observed in the TempO-Seq data (Fig 2G). Only *S100A7A* had a log fold change difference between males and females in the same direction as the TempO-Seq data. Notably, the Schartl dataset included deeper CM samples and the sequencing data was derived from RNA-Seq data rather than TempO-Seq, which may contribute to the sex differences in differential expression between BN and CM. However, overall, we observe consistency in the gene changes between BN and CM across the majority of the genes assessed from the TempO-Seq and Schartl datasets. Since the few gene expression differences observed between sexes were not consistent across datasets, there is limited evidence to suggest a pervasive impact of sex on gene changes through melanoma progression.

## Predominant changes in immune pathways between BN and CM

Next, a pathway enrichment analysis was performed across the union of all genes identified in males, females, or both. Genes were restricted for inclusion based on a log fold change of at least 1.5. Pathway enrichment analysis identified enrichment primarily in immune-related pathways: cytokine signaling, interferon signaling, interleukin-10 signaling,

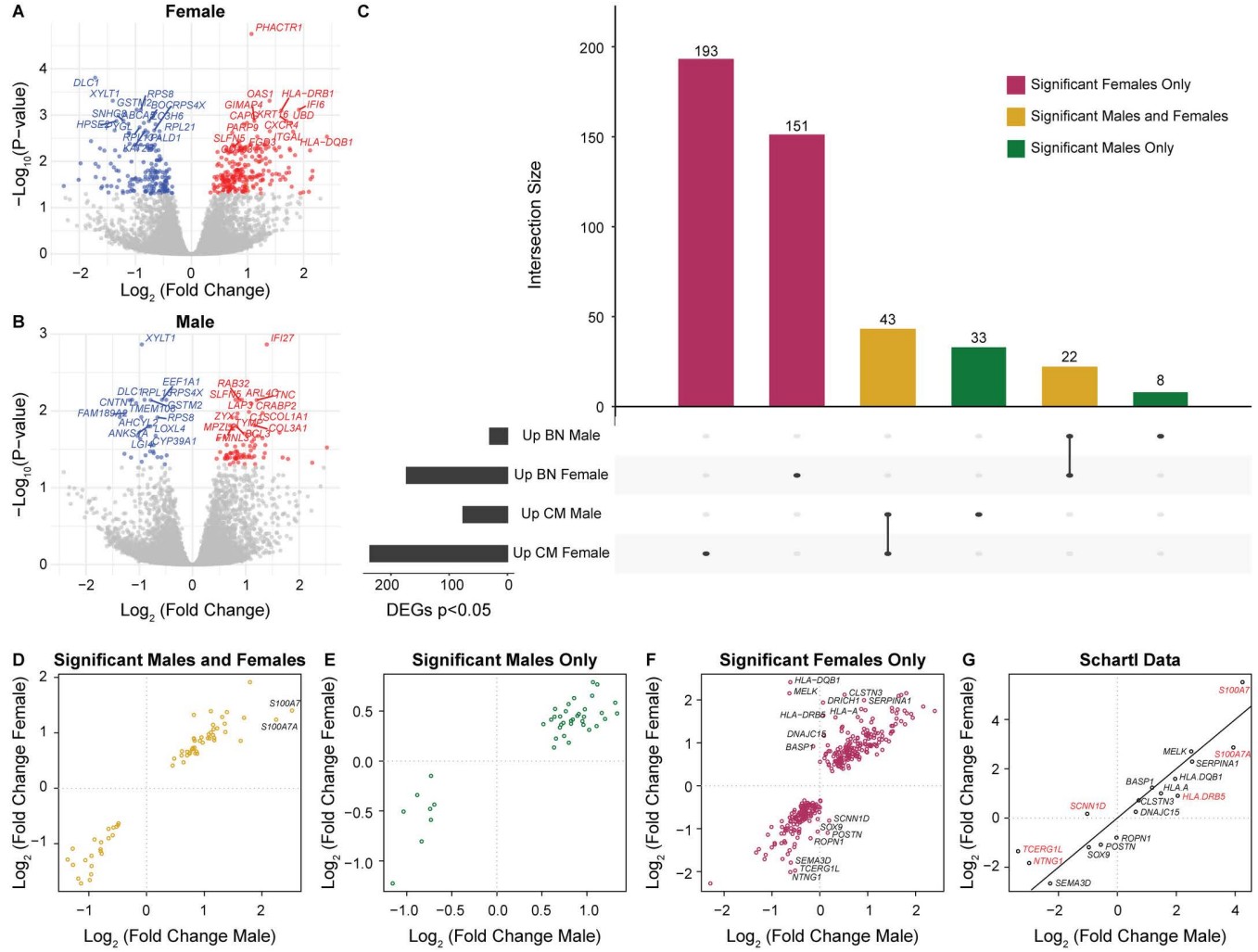

**Fig 2. Differentially expressed genes between benign nevi (BN) and cutaneous melanoma (CM) are consistent in males and females. A-B)** Volcano plot of differentially expressed genes identified in the TempO-Seq data between BN and CM in **(A)** females and **(B)** males. Genes that are significantly (Benjamini-Hochberg adjusted p<0.05) upregulated in CM are shown in red dots, and genes that are upregulated in BN are shown in blue dots. The top 15 genes upregulated in CM and BN are annotated on the plot. **C)** Upset plot of differentially expressed genes in males and females. Genes shared between both males and females are colored yellow, genes unique to females are colored pink, and genes unique to males are colored green. **D-F)** Comparisons of the log fold change of differentially expressed genes in females and males. Genes with a difference in the log fold change (absolute value of the log fold change in males minus the log fold change in females) greater than one between males and females are annotated (range 1.01-3.03). **(D)** Genes significant in both males and females. **(E)** Genes significant in only males. **(F)** Genes significant in only females. **(G)** Comparison of the BN to CM log fold change in gene expression between males and females in the Schartl dataset. Genes are visualized if they had a difference in the log fold change greater than one between males and females in the TempO-Seq data. Genes are colored red if they have a difference in the log fold change greater than one between males and females in the Schartl data.

adaptive immune system, major histocompatibility complex class II antigen presentation, T cell receptor signaling, immunoregulatory interactions between lymphoid and non-lymphoid cells, neutrophil degranulation, and metal sequestration by antimicrobial proteins (Fig 3). Notably, of the genes included in these enriched pathways, many are consistent with previous differential expression results in published analyses. *SELL*, *CXCL1*, *LCP2*, and *S100A7* all showed consistent gene expression differences between BN and CM in at least one dataset previously analyzed by our group [39]. Furthermore, comparison with published analyses of differentially expressed genes between BN and CM in the Badal dataset revealed

**Table 2. Summary of datasets used. BN, benign nevi; CM, cutaneous melanoma; DN, dysplastic nevi; FFPE, formalin-fixed, paraffin-embedded; PN, progressing nevi; RNA-Seq, RNA sequencing.**

| Lesion Types (n) | Sex | Breslow thickness | Sample Type | Relationship | Data Type | Data Access | References |
|---|---|---|---|---|---|---|---|
| BN (54) | 54 females 0 males | Not applicable | Fresh samples snap-frozen or stored in RNA*later* | Independent samples | RNA-Seq | GEO GSE98394 | **Badal** et al. 2017 [32] |
| CM (102) | 40 females 62 males | 0.24-34 mm | | | | | |
| BN (23) | 8 females 13 males 2 not reported | Not applicable | Laser-capture microdissection of fresh biopsies | Independent samples | RNA-Seq | GEO GSE112509 | **Schartl:** Kunz et al. [33] |
| CM (57) | 27 females 29 males 1 not reported | 0.25-10.5 mm | | | | | |
| PN (17) | 6 females 11 males | Not applicable | Manual microdissection of FFPE sections | Paired lesions | RNA-Seq | phs001550. v2.pl | **Bastian:** Shain et al., 2015 [34] Shain et al., 2018 [35] |
| CM (20) | 9 females 11 males | Not available | | | | | |
| BN (5), DN (7) and CM (16) | Not utilized | 0.4-6.7 mm | Frozen sections | Independent samples | 1-channel microarray | GEO GSE114445 | **Krueger:** Mitsui et al., 2016 [36] Yan et al., 2019 [37] |
| BN (18), DN (11), CM Clark level I and II (8), CM Clark level III, IV, V (15) | Not utilized | Not available | Fresh biopsy, Clark level III, IV, V CM limited to dermal portion | Independent samples | 2-channel microarray | GEO GSE12391 | **Scatolini** et al. 2010 [38] |

overlap with consistent directionality in 15 of the 19 genes (*IFI6, USP18, DLC1, HLA-DRB1, HLA-DRB5, HLA-A, INPP5D, CXCL1, LCP2, CD3E, CD3G, SELL, ITGAL, SERPINA1,* and *S100A7*) [32], supporting the biological relevance of these genes in melanoma progression.

## PN cluster in a component distinct and intermediate to BN and CM

Since the batch effect identified between BN/CM and DN was completely confounded with sample type, we could not move forward with the creation of a model that distinguished intermediate lesions with the TempO-Seq data. Therefore, three available RNA-Seq datasets were obtained including data from BN, PN and CM. The Badal [32] and Schartl [33] datasets contain RNA-Seq data from BN and CM (Table 2). The Bastian dataset [35,34] contains RNA-Seq data from CM in association with a nevus, macrodissected into CM and PN regions (Table 2). PCA analysis demonstrated a profound batch effect between the three datasets (Fig 4A). This batch effect was removed with standard statistical approaches, described in the methods (Fig 4B). After the batch correction, PN clustered in a component distinct and intermediate to BN and CM.

## Transcriptomic biomarker distinguishes stages of melanoma progression

After demonstrating that PN occupy an intermediate state between BN and CM, we created a transcriptomic biomarker to distinguish the three lesion types (Fig 5). 1000 iterations of a regularized regression model were performed across the three lesion types, and the genes selected and their coefficients were tracked for each iteration. Twenty-three genes were selected across 100% of the regularized regression models: *PYGL, AP000845.1, PHYHIP, WSCD1, FBXO7, TRPM1, SLC4A4, NALCN, FRMD4B, HHATL, COL1A1, CRYM, EPOP, RGS1, KRT6C, IGHG1, CNTN1, MMP11, GZMM,*

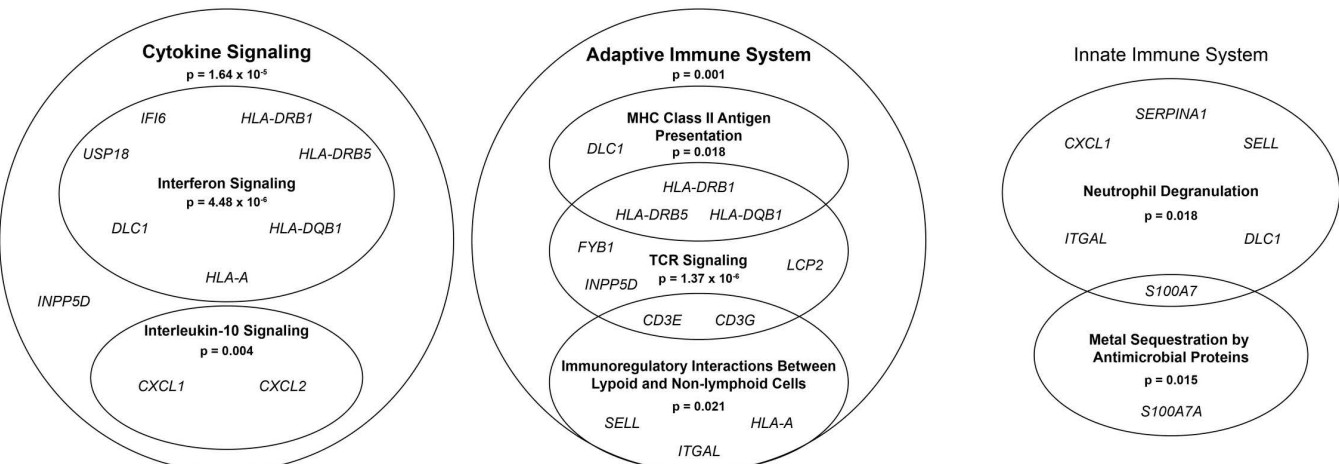

**Fig 3. Predominant changes in immune pathways between benign nevi (BN) and cutaneous melanoma (CM).** Pathway enrichment analysis was performed with Reactome's over-representation analysis on genes identified as differentially expressed between BN and CM with a Benjamini-Hochberg-adjusted $p < 0.05$ and a log fold change $> 1.5$ in either males and/or females. Pathways were considered enriched if they had a false discovery rate-adjusted p-value $< 0.05$. The significantly enriched pathways are represented as circles, with nested circles indicating pathways that are a subset of the higher pathway. Cytokine signaling, adaptive immune system, and innate immune system pathways are shown separately. Bolded pathway names are significantly enriched. Cytokine signaling and adaptive immune system were significantly enriched pathways. The innate immune system pathway was not significantly enriched but is shown as the higher pathway encompassing significantly enriched pathways – neutrophil degranulation and metal sequestration by antimicrobial proteins. Within T cell receptor (TCR) signaling, the generation of second messenger molecules ($p = 9.05 \times 10^{-8}$), translocation of ZAP-70 to immunological synapse ($p = 4.48 \times 10^{-6}$), phosphorylation of CD3 and TCR ζ chains ($4.48 \times 10^{-6}$), regulation of T cell activation by CD28 family ($p = 4.06 \times 10^{-4}$), and co-inhibition by PD-1 ($p = 4.48 \times 10^{-6}$) were also significant but were not diagramed. All significant pathways fell under the larger category of the immune system ($p = 2.95 \times 10^{-6}$), which was also significantly enriched. MHC, major histocompatibility complex.

*AP001880.1, TTYH3, TMEM132A,* and *PRAME* (Fig 5A). We then fit a linear regression model for these 23 genes (Fig 5B) and tested the performance in the training data. Consistent with the original PCA of the data, a PCA restricted to the genes included in the transcriptomic biomarker shows a clear separation of BN and CM with PN occupying an intermediate state (Fig 5C). When we compared the predicted scores for the three lesion types (Fig 5D), BN vs. PN had the lowest area under the receiver operator characteristics curve (AUC) of 0.930, PN vs. CM was next with an AUC of 0.998, and BN vs. CM had the highest AUC at 0.9997. Therefore, the model demonstrated high performance in classifying BN, PN, and CM in the training dataset.

### Genes from the transcriptomic biomarker show consistent directionality across platforms

Subsequently, we tested the consistency of the model in the TempO-Seq data and two additional, independent microarray datasets with BN, DN, and CM samples provided by Krueger [36,37] and Scatolini [38] (Table 2). Since the transcriptomic biomarker could not be directly tested due to the differences in technology, linear regression models were fit in each of the datasets independently. Of the 23 genes selected in the original model, consistent directionality and magnitude were observed in twelve of the genes in the TempO-Seq data, six genes in the Kruger data, and eleven genes in the Scatolini data (Fig 6). Consistent directionality was observed in three additional genes in the TempO-Seq data, three additional genes in the Krueger data, and two additional genes in the Scatolini data. Finally, overlapping confidence intervals were observed in three genes in the TempO-Seq data, five genes in the Krueger data, and two genes in the Scatolini data. All remaining genes were either absent from the dataset or inconsistent in both magnitude and directionality. Notably, for three genes, *MMP11, TRPM1,* and *CNTN1,* two probes were present in TempO-Seq, and both are considered separately. If only the more consistent probe is considered, there is evidence for consistency in 18/20 (90%) of the selected genes

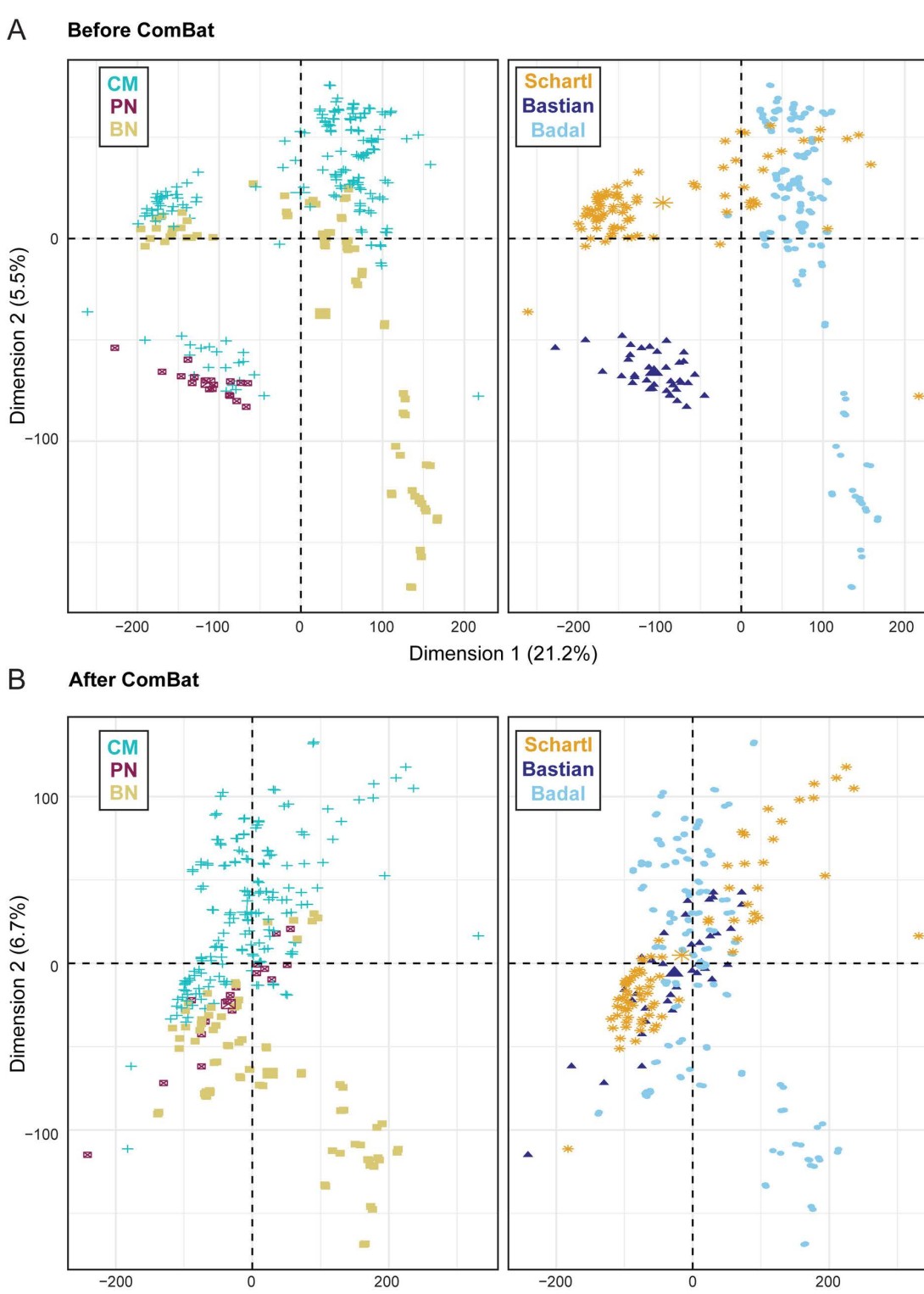

**Fig 4. Progressing nevi (PN) clustered in a component distinct and intermediate to benign nevi (BN) and cutaneous melanoma (CM).** Principal component analysis (PCA) on expression data from three independent RNA sequencing (RNA-Seq) datasets from BN, PN, and CM. **A)** PCA analysis was performed on centered log-ratio transformed RNA-Seq data. Colored by lesion type (left) and dataset (right). **B)** PCA analysis was performed on centered log-ratio transformed RNA-Seq data after adjustment for the batch effect from the three different datasets using the ComBat package in R. Colored by lesion type (left) and dataset (right).

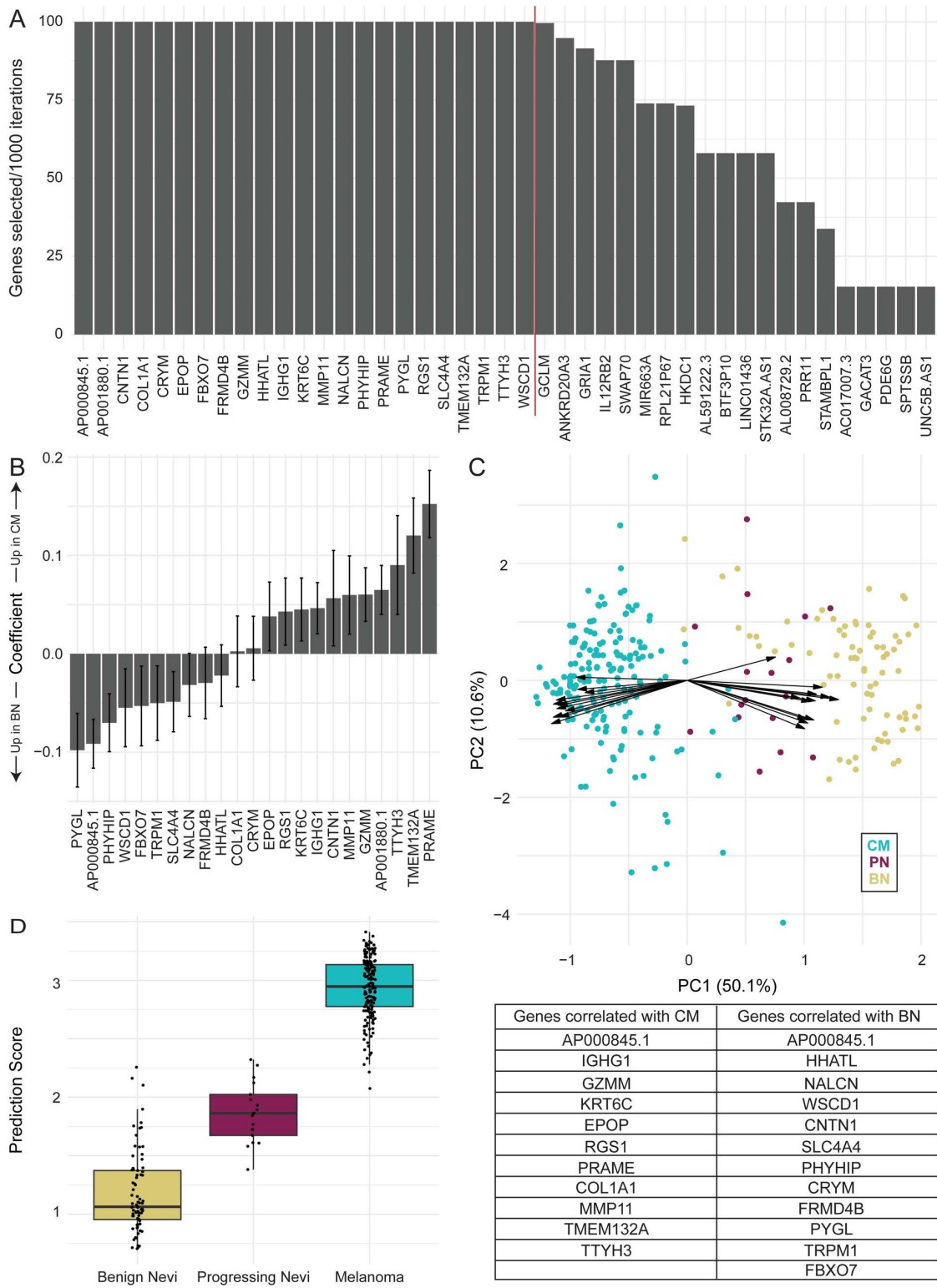

**Fig 5. Transcriptomic biomarker distinguishes stages of melanoma progression. A)** Genes selected across 1000 iterations of ordinal regularized regression models with benign nevi (BN), progressing nevi (PN), and cutaneous melanoma (CM) assigned the values of 1, 2, and 3, respectively. For visualization, only genes selected in at least 10% of the iterations are included. The 23 genes to the left of the red line were selected 100% of the time

and included in the putative transcriptomic biomarker. **B)** Coefficients fit for each of the genes selected for inclusion in the transcriptomic biomarker. Coefficients estimated with an ordinal linear regression model. Error bars represent the standard error calculated for the linear regression model. **C)** Biplot of the BN, PN, and CM samples with the 23 genes selected for the transcriptomic biomarker. Arrows indicate the direction and magnitude of the effect of each gene in separating the lesion types. The table below lists the genes corresponding to the arrows from top to bottom. **D)** Prediction scores for each lesion type with the 23 gene transcriptomic biomarker.

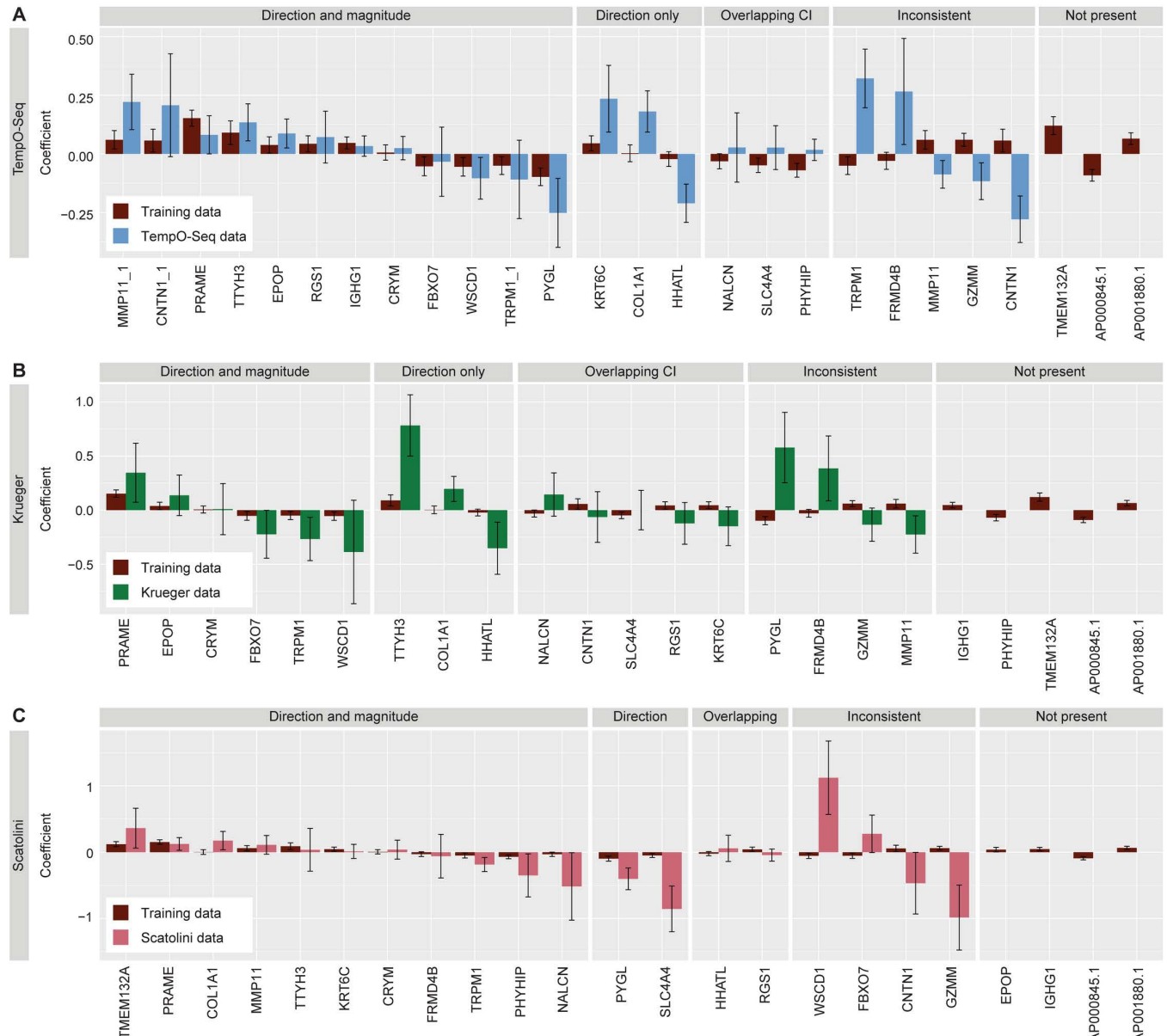

**Fig 6. Consistent magnitude and directionality of coefficients for selected biomarker genes in independent datasets.** Regression models were fit using the same 23 genes in three independent datasets and the coefficients were compared to the coefficients from the original model. Genes are sorted into panels based on the consistency of the coefficients: 1) consistent directionality and overlapping confidence intervals (CI), 2) consistent directionality and non-overlapping confidence intervals, 3) inconsistent directionality and overlapping confidence intervals, 4) inconsistent directionality and non-overlapping confidence intervals, and 4) genes not present in the targeted sequencing. Comparison of coefficients from the original model with coefficients fit in the **A)** TempO-Seq data, **B)** Krueger data, and **C)** Scatolini data.

present in the TempO-Seq data. Similarly, 14/18 genes (78%) show evidence of consistency in the Krueger dataset, and 15/19 genes (79%) show evidence of consistency in the Scatolini data. Notably, *PRAME, CRYM, TRPM1, COL1A1* and *TTYH3* show consistent directionality only or magnitude and directionality across all three datasets, whereas *GZMM* shows no consistency in any of the three test datasets. Overall, these results demonstrate the predictive potential of the selected genes as they show consistent differences between nevi and CM neoplasms across datasets and technologies.

## Immunohistochemical staining for MMP11 and PYGL in BN and CM reveals changes consistent with the transcriptomic biomarker

To further investigate the utility of the genes selected in the transcriptomic biomarker for use in distinguishing CM from BN, we tested two with IHC staining. MMP11 and PYGL were selected as these genes both demonstrated high, consistent coefficients in our transcriptomic modeling and had antibodies validated through the Human Protein Atlas (https://www.proteinatlas.org/). MMP11 and PGYL staining was scored in nevus melanocytes and melanoma cells; both MMP11 and PGYL displayed cytoplasmic staining.

The most common pattern for MMP11 staining within benign nevi was a staining gradient in dermal nevus melanocytes, with more intense staining in the superficial dermis and gradual loss of intensity towards the deeper dermis (Fig 7A). Other BN samples showed a uniform staining pattern (Fig 7B). In contrast, the staining gradient was rarely observed in CM, which predominantly displayed a uniform staining pattern (Fig 7C and 7D). On quantification, no differences were observed in either the staining percent or intensity of staining in the epidermis or dermis of the BN and CM samples (Fig 7E and 7F). However, when the patterns of staining were categorized, the dermis of BN samples had a significant increase in the staining gradient relative to the dermis of CM samples (Fig 7G). The gradual loss of MMP11 staining in the dermis of BN may explain the lower overall level of *MMP11* gene expression in BN in RNA-Seq analysis.

PYGL staining showed uniform strong staining of nevus melanocytes in the dermis of BN (Fig 7H and 7I). By contrast, little to no staining was observed within melanoma cells in CM (Fig 7J and 7K). On quantification, the percentage of nevus melanocytes/melanoma cells stained was significantly higher for BN compared to CM in both the epidermis and dermis (Fig 7L). Staining intensity was also significantly higher for BN compared to CM in the dermal portions of the lesions, though the epidermal portions did not reach significance (Fig 7M). The dermal portions of BN showed significantly greater staining intensity compared to the epidermal portions of BN (Fig 7M). In terms of the observed staining patterns, both the epidermal and dermal portions of the BN showed different staining patterns from the epidermal and dermal patterns in the CM samples, predominantly driven by the lack of staining in most CM samples (Fig 7N). The dermal portions of the BN showed uniformly strong staining across all samples, which also differed from the epidermal portions of the BN lesions which more frequently had no staining or moderate staining (Fig 7N). These results are consistent with the lower RNA-Seq *PYGL* gene expression in CM compared to BN.

## Discussion

While the original intent of this study was to build a transcriptomic biomarker to distinguish BN, DN, and CM in the newly acquired TempO-Seq data, the batch effects likely caused by the length of the FFPE specimen storage prevented this approach. Although other demographic characteristics differed between the three groups, only the storage length of the samples is consistent with the complete separation of the BN/CM samples from the DN samples on principal components analysis. This shortfall draws attention to the need to consider the length of specimen storage when creating a cohort for sequencing studies. Many years of experience in biomarker research have identified a set of best practices for risk and prediction biomarkers [40–42]. In addition to the length of storage of any biospecimens, these best practices consider the sampling scheme under which the biospecimens were obtained and how well these match the intended use of the biomarker. Simon et al. [41] discusses these best practices from the perspective of the level of evidence. Specifically, some biospecimen collection and study designs necessitate more validation than others. For example, samples from controlled

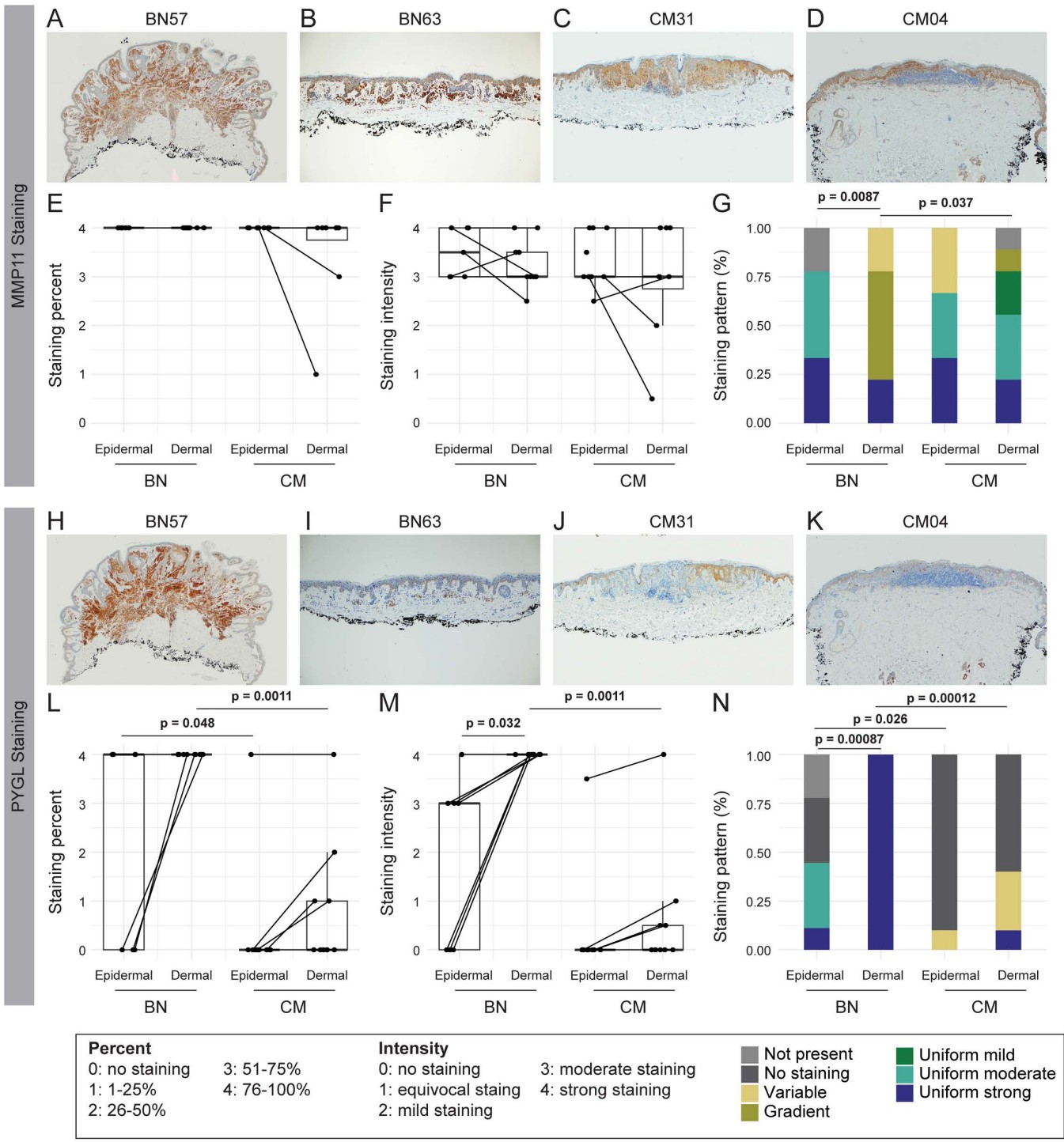

**Percent**
0: no staining 3: 51-75%
1: 1-25% 4: 76-100%
2: 26-50%

**Intensity**
0: no staining 3: moderate staining
1: equivocal staing 4: strong staining
2: mild staining

Not present Uniform mild
No staining Uniform moderate
Variable Uniform strong
Gradient

**Fig 7. Immunohistochemical (IHC) staining for MMP11 and PYGL reveals changes consistent with the transcriptomic biomarker. (A-D)** Representative MMP11 IHC staining in two benign nevi (BN) and two cutaneous melanoma (CM) samples. **(A)** BN with uniform strong staining of nevus melanocytes in the epidermis and a staining gradient in the dermis with a progressive decrease in staining deeper in the dermis. **(B)** BN with uniform strong staining of nevus melanocytes in both the epidermis and dermis. **(C)** CM with uniform moderate staining of melanoma cells in both the epidermis and dermis. **(D)** CM with uniform strong staining of melanoma cells in both the epidermis and dermis. **(E)** Frequency of MMP11 staining in the epidermal and dermal components of BN and CM samples. **(F)** Intensity of MMP11 staining in the epidermal and dermal components of BN and CM samples. **(G)**

Pattern of MMP11 staining categorized as no staining, variable staining, staining gradient, uniform mild, uniform moderate, and uniform strong staining. Not present indicates that the slide did not contain the epidermal or dermal portions for inspection and quantification. (H- K) Representative PYGL IHC staining in two BN and two CM samples. (H) BN with uniform strong staining in nevus melanocytes in both the epidermis and dermis. (I) BN sample with no staining of nevus melanocytes in the epidermis and uniform strong staining of nevus melanocytes in the dermis. (J) CM sample with no staining in melanoma cells in the epidermis and variable staining of melanoma cells in the dermis. (K) CM sample with no staining of melanoma cells in the epidermis or dermis. Staining in keratinocytes, especially basal keratinocytes, was observed and not included in the scoring. (L) Frequency of PYGL staining in the epidermal and dermal components of BN and CM samples. (M) Intensity of PYGL staining in the epidermal and dermal components of BN and CM samples. (N) Pattern of PYGL staining observed in the samples, groups as in (G). Testing of the differences in the paired epidermal to dermal gene-wise expression was conducted using the Wilcoxon rank sum test statistic. The Wilcoxon rank sum test was also applied for all comparisons of BN to CM samples, where comparisons were limited to either epidermal:epidermal or dermal:dermal comparisons. Fisher's exact test was used to compare the differences in the frequency of each pattern for panels G and N.

clinical trials are a relative gold standard compared to retrospective/observational, often convenience samples that have a lower level of evidence. Having clearly written standard operating procedures that outline sample processing through sequencing is described as a must-have for reproducibility and biomarker validation as well as the demonstration of clinical utility. While appropriate statistics or analytics are helpful, many times it is not the direct cause of the lack of reproducibility of biomarker development efforts. This current project demonstrates the impact of both sampling and sample quality on biomarker development, even when documentation, sample preparation, and analysis best practices are followed. Critically, new sample acquisition should prioritize inclusion of DN samples, since these are conspicuously absent from the available RNA-Seq data and are the neoplasms that present the greatest diagnostic challenges [4–6].

Despite the limitation in the use of the DN samples, the comparison of BN and CM from the TempO-Seq data was employed to probe biological changes through melanoma progression. Differential expression analyses were performed with age-adjusted values and were segregated by sex to account for the underlying differences in the age and sex distributions of the samples. Overall, we identified 450 differentially expressed genes between BN and CM detected in either males only, females only, or both sexes, with 269 genes upregulated and 181 genes downregulated in CM. Given sex differences in the incidence and mortality of CM [43], we evaluated the evidence for a sex difference in gene expression. The vast majority of gene expression differences between BN and CM were of similar magnitude and direction in both males and females. While 4% of the differentially expressed genes had a more than one log fold change difference between males and females in our TempO-Seq dataset, the sex differences in the expression of these genes between BN and CM were not substantiated in a second dataset. Thus, there were no substantial differences in the gene expression between sexes. Our results are consistent with a recent study demonstrating that the improved survival in melanoma observed in females is not a direct effect of sex, but rather an indirect effect of clinicopathological features [44].

Among the differentially expressed genes identified between BN and CM, we identified significant enrichment of immune-related pathways, consistent with a previous study by our team [39]. Several of the genes enriched in immune-related pathways (Fig 3) have previously been identified as having functions anticipated to contribute to melanoma initiation and progression. Interferon-α inducible protein 6 (encoded by *IFI6*) regulates NRAS-induced melanoma development and growth, via regulation of *E2F2* expression and DNA replication [45]. *USP18* expression in melanoma cells improves immune-mediated control of *in vivo* tumor growth [46]. *DLC1* promotes melanoma growth and metastasis [47]. *CXCL1* and *CXCL2* promote melanomagenesis [48,49]. *FYB1* is a significantly mutated gene in acral melanoma, but its role in cancer remains unknown [50]. High *LCP2* expression correlates with improved melanoma survival, increased tumor-infiltrating immune cells, and improved response to anti-PD1 [51]. In cutaneous melanoma, high *SERPINA1* expression is correlated with improved survival and increased immune cell infiltrate [52]. *S100A7* is linked to tumor growth and metastasis in breast and oral cancer [53]. Additionally, we found overlap of the differentially expressed genes identified in the TempO-Seq dataset with those used in the MyPath Melanoma gene expression profile test used to aid in the diagnosis of ambiguous melanocytic neoplasms [9,10,12]. The overlapping genes, *S100A7*, *CXCL9*, *IRF1*, *LCP2*, and *SELL*, are

implicated in the immune system. Notably, we found that *PRAME*, which has previously been identified as differentially expressed in datasets across technologies [39], was not identified as differentially expressed in TempO-Seq data.

Finally, we created a putative biomarker to distinguish melanoma from BN and PN neoplasms. Within the training data, we demonstrated high performance in distinguishing the stages of melanoma development. Additionally, we demonstrated 78–90% consistency in magnitude or directionality of the coefficients in three independent datasets from different technologies, reaffirming the predictive ability of the genes identified. Our putative transcriptomic biomarker shares the primary component of the MyPath Melanoma test, *PRAME*, but does not share any other genes with the MyPath Melanoma test. The putative transcriptomic biomarker includes additional genes (*TRPM1*, *PYGL*, *COL1A1*, *RGS1, FBXO7)* with roles in cancer initiation and progression or association with CM prognosis. Loss of *TRPM1* expression is associated with poor survival in CM [54]. *PYGL* is involved in glycogen metabolism, which regulates inflammatory responses and tumorigenesis [55–57]. Type 1 collagen, which is encoded in part by *COL1A1,* promotes CM incidence, invasion, and angiogenesis during CM development [58]. Increased RGS1 protein expression is associated with decreased relapse-free and disease-specific survival in CM progression and prognosis [59]. *FBXO7* has been shown to promote oncogenesis and inhibit apoptosis in other cancer types [60].

IHC staining demonstrated changes in MMP11 and PYGL protein expression consistent with the transcriptomic biomarker. While the percent and intensity of MMP11 staining did not change between BN and CM, there was a staining gradient in the dermis observed significantly more frequently in BN samples, which likely contributed to the decreased *MMP11* gene expression detected in BN samples. The staining gradient of MMP11 is similar to that observed with HMB45, which shows a gradient of staining in BN samples and not in CM samples [61]. In contrast, PYGL showed significantly increased staining percent and intensity in BN compared to CM, with all BN samples having strong staining at least in the dermal portion and only one CM sample having any staining. Thus, the IHC studies provide protein-level validation for an upregulated and downregulated gene in the transcriptomic biomarker.

A limitation of our transcriptomic biomarker is that it was created on a small sample size and may suffer from overfitting. Similarly, the IHC staining was performed on a small subset of samples and thin CM have minimal dermal component for scoring. Additional studies for both testing and confirmation are needed to fully develop an early stage diagnostic biomarker for melanoma Furthermore, while TempO-Seq is valuable for model generation, targeted sequencing, quantitative PCR or IHC may be preferable for clinical application. Despite these limitations, the results here demonstrate the feasibility of a transcriptomic biomarker for distinguishing melanocytic lesions. With additional data, the model presented here might be tested or updated to provide a more clinically relevant model for distinguishing stages of melanoma initiation and progression.

## Materials and methods

### Ethics statement

This retrospective research protocol was approved by the University of Arizona Institutional Review Board. Participants from the prior clinical trial (NCT00841204, "Sulindac in Preventing Melanoma in healthy Participants Who Are at Increased Risk of Melanoma") with AN gave signed written consent for use of specimens in future research. The UA Institutional Review Board approved a waiver of informed consent for the retrospective study of existing BN and CM specimens. A chart review of patients seen at the UA Banner University Medical Center Tucson was performed to identify cases diagnosed with compound nevus without atypia (n = 50) or superficial spreading CM (n = 51) with a thickness of less than or equal to 1.0 mm between 2017 and 2021. The remainder of the data analyzed were from publicly available sources under ethical approval from their respective studies.

### Acquisition and processing of patient samples

Five micron serial sections of the identified cases were cut from FFPE tissue blocks, which were archived in a community pathology lab. Additionally, 5 μm serial sections were cut from FFPE AN tissue blocks from a prior chemoprevention trial

conducted between 2009 and 2010 at the UA and Stanford University [19]. The clinical trial samples were processed and paraffin-embedded at the study institutions and stored at the UA Skin Cancer Prevention Laboratory. Histopathological review of hematoxylin and eosin-stained sections of the clinically AN revealed that 61 were DN and 5 were BN.

## TempO-Seq gene expression analysis

The gene expression data was generated using the TempO-Seq Human Whole Transcriptome Assay, which sequences approximately 20,000 protein-coding genes (BioSpyder Technologies Inc.). The sample preparation methods have been described previously [62]. Briefly, TempO-Seq is an extraction-free method. Regions of interest were annotated by a dermatopathologist (P.S.) on hematoxylin and eosin-stained slides. The corresponding regions of interest were scraped from unstained 5 μm serial sections (targeting a total area of 10–20 mm$^2$) and collected into a lysis reagent with mineral oil. Subsequently, the tissue samples were heated at 80°C for 5 minutes to deparaffinize the tissue. The tissues were further digested using a protease until complete lysis was achieved. An aliquot of the resulting lysate was mixed with a combination of TempO-Seq Detector Oligos, which were subsequently hybridized, cleaned up with an exonuclease, and ligated. The samples were then subjected to PCR amplification using barcoded primers, and the amplified products were pooled into a sequencing library. Purification of the sequencing libraries was performed using the NucleoSpin Gel and PCR Cleanup Kit (Clontech). Finally, the samples were sequenced on an Illumina NovaSeqX platform, producing demultiplexed FASTQ files. The FASTQ files were aligned, and raw counts were quantified using the QuasR (v1.26.0) R package [63]. The raw counts and sample metadata are provided (S1 and S2 Tables).

## Differential expression analysis

To visualize the distribution of the samples, raw counts were standardized using a centered log ratio transformation, and the first two principal components were visualized and colored by lesion and the site where the sample was archived. To detect differentially expressed genes between BN and CM samples, we used a linear regression using the limma package in R. Given the differences in sex distribution between the CM (33% female, 67% male) and BN (67% female, 33% male) samples (p=0.0024, chi-squared test), differential expression analysis was stratified by sex. Raw reads were filtered to include genes with a minimum of 5 reads. Subsequently, the raw data was normalized using a trimmed mean of M-values normalization as implemented by limma, log-transformed, and adjusted for quality using the voomWithQualityWeights function in the limma R package [64,65]. To account for the age differences between BN and CM samples, a linear regression was fit for the voom-normalized expression of each gene compared to the age, and the residuals were calculated. Subsequent differential expression analyses were performed on the age-adjusted residuals. Differential expression analyses were performed for male BN vs. male CM and female BN vs. female CM. Linear models included a variance-covariance adjustment for the within-patient sampling assuming a common correlation across genes [66], applied with the duplicatecorrelation function. We used Benjamini-Hochberg false discovery rate (FDR)-adjusted p-values at the 0.05 level of significance. Upset plots were generated using the UpSetR package [67].

To test for consistency in the sex differences in expression, we performed a differential expression analysis between the BN and CM samples from the Schartl dataset [33]. The differential expression analysis was performed in the same manner as outlined above for the TempO-Seq data. Raw reads were filtered to include genes with a minimum of 5 reads, normalized with a trimmed mean of M-values normalization, log-transformed and adjusted for quality using the voomWithQualityWeights function, and age-normalized as described above. The BN to CM log fold changes for genes identified as sex differentially expressed in the TempO-Seq data were compared between males and females.

## Pathway enrichment analysis

Genes identified as significantly differentially expressed (FDR-adjusted p < 0.05 and log fold change > 1.5 for the trimmed mean of M-values normalized, log-transformed, age-adjusted expression) in males, females, or both were analyzed with

a Reactome over-representation analysis. Reactome performs a hypergeometric distribution analysis to determine if the genes identified in a pathway exceed what would be expected by chance. Pathways were considered significantly enriched if they had an FDR-adjusted p-value less than 0.05. A visualization of the enriched pathways and genes within those pathways was created with Adobe Illustrator.

### Acquisition and processing of publicly available datasets

Five publicly available datasets with expression data from BN, DN, PN, and CM were obtained (data accession numbers listed in Table 2). The Badal dataset [32] contains RNA-Seq data from fresh frozen samples from BN and CM and the Schartl dataset [33] contains RNA-Seq data from fresh frozen samples from BN and CM with laser-capture microdissection. The Bastian dataset [35,34] contains RNA-Seq data from FFPE samples of CM in association with a nevus. The samples were manually microdissected into the nevus and melanoma portion with a scalpel under a dissecting microscope. The nevus portion of these lesions is labeled as PN, since these nevi progressed to CM. The Krueger dataset [36,37] contains 1-channel microarray data from fresh frozen samples of BN, DN, and CM. Finally, the Scatolini dataset [38] contains 2-channel microarray data from fresh frozen samples of BN, DN, and CM. While Breslow thickness was not provided for these samples, the CM samples were divided into radial growth phase melanoma (Clark level I and II) and vertical growth phase melanoma (Clark level III, IV, and V).

### Read count quantification

The RNA-Seq data from the Badal, Schartl, and Bastian datasets were obtained as raw FASTQ files and visualized for quality using the FastQC version 0.11.5 (https://www.bioinformatics.babraham.ac.uk/projects/fastqc/). Trimming was performed using TRIMMOMATIC IlluminaClip with the following parameters: seed mismatches 2, palindrome clip threshold 30, simple clip threshold 10, leading quality value 10, trailing quality value 10, sliding window size 4, minimum window quality 15, and minimum read length of 50 [68]. Read mapping was performed with Salmon version 0.11.3 [69]. To account for mismapping of short sequencing reads from X and Y chromosomes, female samples were mapped to the GENCODE GRCh38 genome with all Y chromosome genes hard-masked, and male samples were mapped to the GENCODE GRCh38 genome with the pseudo-autosomal regions of the Y chromosome hard-masked. Raw reads were used for all downstream analyses. Transcripts for the same genes were summed to allow comparison of gene-level expression.

   Both microarray datasets (Krueger and Scatolini) were obtained directly from the prior manuscripts [36–38] as pre-processed expression values as previously described [39].

### Creation of the transcriptomic biomarker

A putative transcriptomic biomarker was created using the combined RNA-Seq data using the Badal, Schartl, and Bastian datasets. On PCA visualization, a profound batch effect was identified between the Badal, Schartl, and Bastian datasets. Therefore, we applied an empirical Bayes framework to adjust for batch effects, implemented with the *ComBat* function from the sva v.3.50.0 package in R [70]. Briefly, expression was standardized across the three datasets, and the mean expression was pooled by the application of a shrinkage estimator using empirical Bayes with parametric priors. Visualization with PCA was repeated after batch effect adjustment. Lesion type was the outcome variable treated as an ordinal variable, with BN = 1, PN = 2, and CM = 3 and genes were potential predictors. Elastic net regularized regression modeling was used for variable selection, using a shrinkage estimator, resulting in an estimated set of coefficients for the genes that optimize the lesion subset classes. This analysis was performed using the glmnet v.4.1-8 package in R [71–73]. Cross-validation of the entire regularized regression model was performed with 1000 bootstrap resamples and the genes selected, their coefficients, and the shrinkage parameter were tracked for each iteration. Shrinkage estimates ranged from 0.073 to 0.15 with a mean of 0.11 across the 1000 resamples. Genes selected across 100% of the regularized regression iterations were used in a final linear regression model.

## Testing of the transcriptomic biomarker

Direct testing of the transcriptomic biomarker in the datasets from Krueger, Scatolini, and the new TempO-Seq dataset was not feasible given the different technologies used to quantify gene expression. Therefore, to test the consistency of the gene changes across datasets and technology, we fit ordinal linear regressions with the genes selected by the putative transcriptomic biomarker in the Krueger [36,37], Scatolini [38], and TempO-Seq data. In the Krueger and Scatolini datasets, BN, DN, and CM were all included and assigned values of 1, 2, and 3, respectively. Since only BN and CM were included in the TempO-Seq data, they were assigned values of 1 and 3, respectively. We, then, visualized the coefficients and their standard errors with paired boxplots and annotated them in one of four categories 1) same directionality and magnitude (overlapping confidence intervals), 2) same directionality with different magnitude (no overlap in confidence intervals), 3) different directionality but overlapping confidence intervals, and 4) opposite directionality and no overlap in confidence intervals.

## Immunohistochemical staining

FFPE samples were selected for 10 BN and 10 CM samples matched on sex, age within 2 years, and skin site. Subsequently, 5 µm sections were cut and mounted on glass slides. IHC staining was performed using automated protocols on a Leica Bond RXm research stainer (Leica Biosystems). Heat induced epitope retrieval was performed using Bond Epitope Retrieval Solution 1 for MMP11 or Bond Epitope Retrieval Solution 2 for PYGL (Leica Biosystems). Sections were stained with rabbit monoclonal anti-MMP11 antibody at a 1:2000 dilution (Invitrogen, cat#MA5–32285) or rabbit polyclonal anti-PYGL at a 1:200 dilution (Atlas Antibodies, cat#HPA000962), followed by detection using the Bond Polymer Refine Detection kit and hematoxylin counterstain. Staining without the primary antibody was utilized as a negative control, and no staining was detected in the specimens. For PYGL, liver tissue was used as a positive control, and for MMP11, placenta and spleen were used as a positive control.

Cell types were identified based on the morphological and histological characteristics. MMP11 staining was identified in melanocytes, keratinocytes, follicular epithelium, pericytes, and smooth muscle cells. MMP11 staining was stronger in melanocytes than keratinocytes. PYGL staining was identified in melanocytes, arrector pila muscle, and cells lining vessels (endothelial cells and/or pericytes), and some basal keratinocytes. Most keratinocytes exhibited no staining or equivocal staining. MMP11 and PYGL staining in nevus melanocytes and melanoma cells on each section was scored with light microscopy by a board-certified dermatopathologist (D.V.S.) and dermatologist (K.T.H.), with agreement reached in each case. The epidermal (intraepithelial)/in situ and dermal/invasive components were scored separately in each case for the percentage of cells stained, staining intensity, and staining pattern. The percentage of cells stained was scored as 0 (no staining), 1 (1–25% of cells), 2 (26–50% of cells), 3 (51–75% of cells), or 4 (76–100% of cells). Staining intensity was scored as 0 (no staining), 1 (blush or equivocal staining), 2 (mild or faint staining), 3 (moderate staining), 4 (strong or intense staining). The staining pattern or distribution of stained cells was categorized as no staining, variable staining (patchy groups of cells in a scattered or random distribution), diffuse staining with a gradient, diffuse mild staining, diffuse moderate staining, or diffuse strong staining. Samples were excluded if they had insufficient BN or CM components for evaluation. The final sample sizes were n = 9 BN and n = 9 CM for MMP11 and n = 9 and n = 10 for PYGL. Diagnostic histologic features are evident on evaluation of sections and preclude blinding. Photomicrographs were acquired using an Olympus BX53F microscope and Olympus DP74 camera.

## Statistical analyses

All statistical analyses were performed in R v4.3.2. The statistical tests used are indicated in the figure legends or text. Non-parametric tests were applied to avoid assumptions of normality. The threshold for significance was set to $p < 0.05$.

## Supporting information

**S1 Table. Full TempO-seq expression data.**
(CSV)

**S2 Table. Full metadata for TempO-seq dataset.**
(CSV)

## Acknowledgments

We thank Mary Krutzsch for technical assistance in the preparation of RNA from the patient specimens. We thank Susan M. Swetter from Stanford University for providing atypical nevi specimens and for her critical review of the manuscript. We thank Anne Macy for proofreading and editing the manuscript. We acknowledge Research Computing at Arizona State University for providing high-performance computing resources that have contributed to the research results reported in this paper. Immunohistochemical staining was performed by the University of Arizona Cancer Center Tissue Acquisition and Cellular/Molecular Analysis Shared Resource.

## Author contributions

**Conceptualization:** Clara Curiel-Lewandrowski, H-H Sherry Chow, Bonnie J. LaFleur, Karen Taraszka Hastings.

**Data curation:** Elizabeth Borden, Michael Yozwiak.

**Formal analysis:** Elizabeth Borden, Melissa A. Wilson, Bonnie J. LaFleur.

**Funding acquisition:** Clara Curiel-Lewandrowski, H-H Sherry Chow, Bonnie J. LaFleur, Karen Taraszka Hastings.

**Investigation:** Elizabeth Borden, Colin T. Hastings, Nithish Prakash, Tyler Kuo, Danielle Vargas de Stefano, Karen Taraszka Hastings.

**Methodology:** Elizabeth Borden, Melissa A. Wilson, Bonnie J. LaFleur, Karen Taraszka Hastings.

**Project administration:** H-H Sherry Chow, Karen Taraszka Hastings.

**Resources:** Edgar Tapia, Paul Sagerman, Kenneth H. Buetow, Melissa A. Wilson, Clara Curiel-Lewandrowski, H-H Sherry Chow.

**Supervision:** Kenneth H. Buetow, Melissa A. Wilson, Bonnie J. LaFleur, Karen Taraszka Hastings.

**Visualization:** Elizabeth Borden, Colin T. Hastings, Nithish Prakash, Tyler Kuo.

**Writing – original draft:** Elizabeth Borden, Colin T. Hastings, Nithish Prakash, Tyler Kuo, Karen Taraszka Hastings.

**Writing – review & editing:** Elizabeth Borden, Melissa A. Wilson, Clara Curiel-Lewandrowski, H-H Sherry Chow, Bonnie J. LaFleur, Karen Taraszka Hastings.

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
