## [Decision Letter · Decision Letter 0]

8 May 2025

PGENETICS-D-25-00149

Towards a transcriptomic biomarker for the classification of melanocytic neoplasms

PLOS Genetics

Dear Dr. Hastings,

Thank you for submitting your manuscript to PLOS Genetics. After careful consideration, we feel that it has merit but does not fully meet PLOS Genetics's publication criteria as it currently stands. Therefore, we invite you to submit a revised version of the manuscript that addresses the points raised during the review process.

Please submit your revised manuscript within 30 days Jun 07 2025 11:59PM. If you will need more time than this to complete your revisions, please reply to this message or contact the journal office at plosgenetics@plos.org. Please include the following items when submitting your revised manuscript:

We look forward to receiving your revised manuscript.

Kind regards,

Shuguo Sun

Guest Editor

PLOS Genetics

Hongbin Ji

Section Editor

PLOS Genetics

Aimée Dudley

Editor-in-Chief

PLOS Genetics

Anne Goriely

Editor-in-Chief

PLOS Genetics

**Journal Requirements:**

At this stage, the following Authors/Authors require contributions: Elizabeth S. Borden, Colin T. Hastings, Nithish Prakash, Tyler Kuo, Edgar Tapia, Paul Sagerman, Kenneth H. Buetow, Melissa A. Wilson, Clara Curiel-Lewandrowski, H-H Sherry Chow, Bonnie J. LaFleur, and Karen Taraszka Hastings. Please ensure that the full contributions of each author are acknowledged in the "Add/Edit/Remove Authors" section of our submission form.

The list of CRediT author contributions may be found here: https://journals.plos.org/plosgenetics/s/authorship#loc-author-contributions

https://journals.plos.org/plosgenetics/s/submission-guidelines#loc-parts-of-a-submission

- ® on page: 3.

5) Thank you for stating " The Bastian dataset is available through the database of Genotypes and Phenotypes (dbGaP) with accession number phs001550.v2.pl." We couldn't access the dataset . Please provide a direct link to access the dataset.

**Reviewers' comments:**

Reviewer's Responses to Questions

Reviewer #1: This study by Borden and colleagues aims to create a diagnostic tool to better differentiate benign nevi (BN) from malignant cutaneous melanoma (CM). They created transcriptomic datasets from BN and CM and also included publicly available data in their analysis. The authors find that several genes can be used as markers to distinguish BN from CM in their TempO-Seq dataset that are involved in “immune’/”inflammation” pathways. However, they observe batch effects due to duration of sample storage and see differences in gene expression due to gender. Lastly, using publicly available data, they identify a set of genes that may differentiate between CM and BN.

The manuscript is interesting because it investigates transcriptomics as a new diagnostic tool for CM, a malignant condition whose early diagnosis is essential for best outcomes. However, they authors do not perform further studies to confirm their findings (e.g. qPCR, immunohistochemistry, in vitro functional studies), which is a disadvantage of the manuscript. In addition, the differences in gene expression in females and males remain unclear.

Minor points:

•Line 89: You are referring to “from the prior trial”. Can you list, which specific trial you are referring to?

•Table 2: Can you please add the p-values mentioned from the text (line 219-226) to the table?

•Line 256: Are any of the differentially expressed genes in females located on chromosome X and if so, which are they?

•Line 279: Did you investigate other reasons for such significant differences in gene expression between the samples? E.g. the thickness of skin sample? Pre-existing conditions? Age of probands?

•Line 301-304: Please add the rate-adjusted p-value for all listed pathways.

•Line 323: Please explicitly state why you could not use the TempO-Seq data to build a model.

•Did you also test the genes and pathways identified from the TempI-Seq data in the publicly available datasets?

•I would appreciate confirmatory studies (qPCR, immunohistochemistry, in vitro functional studies) to confirm that these values have an impact in cutaneous melanoma.

•Line 396: Are the differences in gene expression cause be storage length or patient gender or both? Please discuss.

•Discussion: What specific confirmational test would you suggest? Do you think that TempO-Seq is a good diagnostic tool for CM?

Reviewer #2: The authors point out the importance of establishing a transcriptomic biomarker to distinguish between the different melanomas, while also identifying the genetic changes occuring during meningiomagenesis. The authors adjust their analysis based on patient sex and age for each type of nevus, followed by comparing their data to the publically available datasets. Albeit some discrepancies, the study holds novelty and should be considered for further evaluation. This reviewer has some minor comments which should be considered by the authors.

1. On line 216, the authors mention "AN". Please expand on the abbreviation as it is not mentioned prior in the text of the manuscript, only in the legend of table 2 as atypical nevi.

2. The authors mention that the CM samples obtained were from significantly older patients relative to samples obtained from patients with BN and AN. Could the authors comment on this before begining the analysis. Is it known that samples from older patients have different transcription profiles relative to transcriptiomic profiles of CM obtained from younger patients? Please cite relavant articles to this note.

3. The authors indicate that there are significant site differences between the different nevi. They should explain if sites need to be comparable for accurate analysis. If the site is indeed of consequence in melanoma, the authors should perform analysis to compare specific sites among the different nevi.

4. The tables should be placed in a chronological order as Table2 is mentioned before Table1.

5. In line 278, where the authors mention "since the gene expression differences between the sexes were not......consistent in males and females.". This sentence is extremely confusing and the authors should consider simplifying it for better understanding.

6. The Log fold changes for the data in figures 2 and 3 is not consistent. The authors look at the differentially expressed genes in CM and BN in the males and females, log change difference cut off is 1, while that for enriched immune pathways is 1.5. Is there a reason for this? To be consistent, the authors might consider including a log change difference cut off of 1.5 or higher for the data in figure2 as well.

7. The authors point out that there is a "clear separation of the three lession types on line 347 when comparing the genes of PN, CM and BN in a PCA plot. The "clear separation" is that for melanoma and benign, but the PN cluster is more spread in regions with overlap in transcripts from melanoma. Please rephrase this rentence to more accurately represent the data in Figure5C.

-Figure 5C is missing a key.

-Since the number of samples for the PN group is significantly lower relative to malanoma and benign, the "cluster" is not necesasrily defined.

Reviewer #3: The manuscript by Borden et al. describes the development of a transcriptomic biomarker to distinguish benign nevi (BN), progressing nevi (PN), and cutaneous melanoma (CM). The study addresses an important clinical need and employs multiple datasets and statistical approaches. Overall, the approach is sound, and the results are promising. However, several areas require clarification and expansion to improve readability, rigor, and reproducibility.

Major Points

1. Introduction:

o Line 53-60: The author could add a few sentences describing the key histopathological features that characterize melanocytic lesions ranging from dysplastic nevi to thin (T1a) cutaneous melanoma to make the text more accessible to non-specialists.

o The author could also discuss whether gender influences disease progression.

2. Clarify Batch‐Effect Removal

o Lines 328–329: The text refers to “standard statistical approaches” for batch‐effect removal but does not specify the method. Please provide a brief description of the statistical approach used for batch-effect removal, as details were not provided in the Materials and Methods section (lines 184–185).”

3. Validation using multiple datasets

o Lines 273–274: The author should use more than one dataset to investigate whether sex differences exist in these 18 genes between BN and CM.

4. Protein‐Level Validation

o Although the transcriptomic biomarker shows consistency across platforms, the functional relevance of these genes would be strengthened by orthogonal (e.g., IHC) validation of at least a subset of these genes at the protein level.

5. Inclusion of Dysplastic Nevi (DN)

o Line 395-410: Dysplastic nevi present the greatest diagnostic challenge, yet DN samples were excluded from the TempO-Seq analyses due to batch confounding. Please discuss, if possible, how future studies might prioritize the collection and processing of DN samples to fill this critical gap.

**Have all data underlying the figures and results presented in the manuscript been provided?**

Reviewer #1: Yes

Reviewer #2: Yes

Reviewer #3: Yes

PLOS authors have the option to publish the peer review history of their article (what does this mean? ). If published, this will include your full peer review and any attached files.

**Do you want your identity to be public for this peer review?** For information about this choice, including consent withdrawal, please see our Privacy Policy .

Reviewer #1: No

Reviewer #2: No

Reviewer #3: No

**Figure resubmission:**
---

## [Decision Letter · Decision Letter 1]

3 Sep 2025

Dear Dr Borden,

We are pleased to inform you that your manuscript entitled "Towards a transcriptomic biomarker for the classification of melanocytic neoplasms" has been editorially accepted for publication in PLOS Genetics. Congratulations!

Yours sincerely,

Shuguo Sun

Guest Editor

PLOS Genetics

Hongbin Ji

Section Editor

PLOS Genetics

Aimée Dudley

Editor-in-Chief

PLOS Genetics

Anne Goriely

Editor-in-Chief

PLOS Genetics

Comments from the reviewers (if applicable):

Reviewer's Responses to Questions

**Comments to the Authors:**

Reviewer #1: Thanks for making the indicated changes.

Reviewer #2: The authors have addressed all the concerns raised by this reviewer and the manuscript should be accepted in Plos Genetics.

Reviewer #3: The authors have addressed most of the concerns raised in the earlier review process.

Key clarifications have been added regarding histopathological context, batch-effect correction, and the potential influence of sex and age on gene expression. Importantly, the revised version now includes immunohistochemical validation of two biomarker genes (MMP11 and PYGL), which strengthens the translational relevance of the findings. The discussion of dysplastic nevi has also been expanded, highlighting a critical area for future investigation.

Overall, the revisions substantially improve the clarity, rigor, and impact of the manuscript.

**Have all data underlying the figures and results presented in the manuscript been provided?**

Reviewer #1: None

Reviewer #2: Yes

Reviewer #3: None

PLOS authors have the option to publish the peer review history of their article (what does this mean? ). If published, this will include your full peer review and any attached files.

**Do you want your identity to be public for this peer review?** For information about this choice, including consent withdrawal, please see our Privacy Policy .

Reviewer #1: No

Reviewer #2: No

Reviewer #3: No

**Data Deposition**

http://datadryad.org/submit?journalID=pgenetics&manu=PGENETICS-D-25-00149R1

**Press Queries**

---

## [Editor Report · Acceptance letter]

PGENETICS-D-25-00149R1

Towards a transcriptomic biomarker for the classification of melanocytic neoplasms

Dear Dr Borden,

We are pleased to inform you that your manuscript entitled " 

Towards a transcriptomic biomarker for the classification of melanocytic neoplasms" has been formally accepted for publication in PLOS Genetics! Your manuscript is now with our production department and you will be notified of the publication date in due course.

With kind regards,

Olena Szabo

PLOS Genetics

On behalf of:
